# Highly Improved Captures of the Diamondback Moth, *Plutella xylostella*, Using Bimodal Traps

**DOI:** 10.3390/insects16090881

**Published:** 2025-08-24

**Authors:** Andrei N. Frolov, Yulia A. Zakharova

**Affiliations:** Laboratory of Agricultural Entomology, All-Russian Institute of Plant Protection, Pushkin, 196608 St. Petersburg, Russia; julia_fefelova@mail.ru

**Keywords:** Lepidoptera, mass trapping, monitoring, pheromone, Plutellidae, sex attractant, superadditive interaction (synergy), UV light-emitting diode (LED)

## Abstract

The diamondback moth, *Plutella xylostella* (L.) (DBM), is highly widespread and the most destructive pest of plants belonging to the Brassicaceae family. Traps equipped with synthetic sex attractant (SSA) of DBM are widely used to monitor the pest population dynamics worldwide. In addition, successful results have been achieved in using SSA-based traps to control DBM populations through mass trapping. However, not only SSA but also light-emitting diodes (LEDs) can be used in traps both for monitoring and for controlling DBM populations, significantly reducing the use of insecticides. We present the results of field trials using traps equipped with SSA and UV LEDs, indicating a powerful synergistic interaction between these attraction means for capturing DBM adults.

## 1. Introduction

The diamondback moth (DBM) *Plutella xylostella* (L.) (Lepidoptera: Plutellidae) is a highly widespread and most dangerous pest of plants from the Brassicaceae family, including broccoli, Brussels sprouts, cabbage, Chinese cabbage, cauliflower, collard, kale, kohlrabi, mustard, radish, and watercress, in Europe, Asia, Africa, America, and Australia [1,2,3,4,5,6,7,8,9], though there is no consensus of the place of the DBM’s origin [10,11,12,13]. In addition to the cultivated, wild-growing, and weedy plants of the Brassicaceae family [1,14,15], the DBM attacks some plants from other botanical families [16,17,18]. Due to its exceptionally high migration activity, the DBM can be found almost anywhere where its host plants grow, and due to climate warming, the harmful activity of this pest is expected to further increase [19,20]. So, the DBM is deservedly considered one of the most difficult insect pests in terms of plant protection and, consequently, one of the most harmful [7,14,21]. For instance, at the end of the 20th century, annual losses from the DBM were estimated at USD 1 billion worldwide [11], and by the end of the first decade of the 21st century, the harm caused by this insect is estimated to be in the range of USD 4–5 billion, according to the most conservative estimates [22].

In Russia, the frequency of DBM outbreaks has greatly increased over the past decade. The geographic range of these outbreaks has also expanded, and they now periodically occur not only in different regions of the European but also in the Asian part of the country (in Siberia), although at the end of the last century, this species was considered only to be a secondary pest capable of causing damage to cabbage crops, only in some years [23,24,25]. The increase in DBM harmfulness in Russia was apparently initiated by the following: (1) climate warming (there is an earlier appearance of the pest in the fields, as well as an increase in the number of generations per season), (2) the expansion of acreage under cabbage crops and primarily under rapeseed, and (3) the spread of minimal tillage, which creates conditions for the accumulation of a wintering stock of the pest [25,26,27]. Thus, in recent years, the harm of the DBM in Russia has increased by several orders of magnitude, and outbreaks of the pest have been observed every 4–6 years. Consequently, this insect is now considered one of the worst pests for cabbage crops in the Russian Federation and its neighboring countries (Belarus, Kazakhstan, etc.) [23].

In Russia, as in other countries, DBM control is based mainly on the use of a wide range of insecticides. Preparations containing active substances from a variety of chemical classes, such as pyrethroids, neonicotinoids, organophosphorus compounds, avermectins, chitin synthesis inhibitors, juvenile hormone analogues, alkaloids, and carbamates, are used, and the effectiveness of many insecticides is ensured by the interaction of two and sometimes even three toxicants [28]. The intensive use of insecticides in Russia, as it is everywhere in the world, leads to breeding resistant DBM populations [23,29]. Currently, 642 cases of resistance have been recorded worldwide in various populations of the DBM to 101 different insecticides [30], and reports of new resistance cases are becoming more frequent in the literature [31,32,33,34,35,36,37]. As a result, the difficulty of controlling the DBM is determined not only by its extremely high migratory activity [11,38,39,40,41,42,43,44,45] but also by its ability to rapidly develop resistance to almost all insecticides, both chemical and biological [1,33,46,47,48,49,50,51]. Therefore, improving DBM management in most countries includes developing and implementing a range of measures, including not only insecticide use but also biological control, host plant resistance, habitat management, regulation of planting period, push–pull strategies, trap cropping, intercropping, crop rotation, genetic control, etc., that try to consolidate into integrated systems [1,3,8,52,53,54,55,56,57,58]. One of the most important elements of these systems is undoubtedly monitoring of the pest [59,60,61,62].

The composition and properties of the female sex pheromone in the DBM have been studied since the 1970s [63,64,65,66,67]. The main components of this semiochemical were first identified as (Z)-11-hexadecenal and (Z)-11-hexadecenyl acetate [68,69], then (Z)-11-hexadecen-1-ol [70]; finally, a minor component, (Z)-9-tetradecenyl acetate [71], was discovered. Later, it was determined that the composition of the DBM sex pheromone varies within the insect’s range, and accordingly, to ensure maximum attractiveness, the composition of the synthetic sex attractant (SSA) for a specific population needs to be clarified [72,73,74,75]. For the DBM found in Russia, the most attractive composition includes (Z)-11-hexadecenal and (Z)-11-hexadecenyl acetate in a ratio of 10:90, with (Z)-11-hexadecen-1-ol added as a minor constituent (0.99% of the total composition) [76].

Traps equipped with DBM SSA are widely used for pest monitoring [60,77,78,79,80,81,82,83,84,85,86,87]. The composition of the SSA, designed for pest monitoring in Russia, has undergone detailed field testing, during which the concentration of active substances and the dispenser type were optimized [76,88]. Currently, two commercial organizations distribute DBM SSA in Russia: JSC Shchelkovo Agrochem (https://betaren.ru/) and Pherotrap LLC (https://pherotrap.ru/).

In addition to monitoring, semiochemicals are increasingly used as a tool for controlling insect pest populations [89]. Thus, to control the DBM, SSA can be used to achieve the following: (1) male disorientation during mating [90,91,92,93,94,95,96,97,98,99], (2) mass capture of adults in order to gain male annihilation [69,100,101,102,103,104,105,106], and (3) the spread of entomopathogenic microorganisms in the pest population by an autodissemination process [107,108,109]. Pilot trials in Russia also have shown promising results using SSA to control the DBM [110].

Although pheromone monitoring is a simple, low-cost, and fairly accurate method for detecting and counting most insect pest species, it is known that sex pheromones in Lepidoptera typically attract only males, while harmful offspring are produced by females [111]. Therefore, it is not surprising that sometimes the density of DBM adults captured by SSA-based traps does not correlate with the density of larvae feeding on plants [60]. For this reason, other attractive semiochemicals are being tried in addition to SSA to lure not only male but also female DBMs. Volatile organic compounds isolated from DBM host plants are among the most attractive to both sexes [112,113,114,115].

Even though DBM adults are capable of flying during the daytime, most of their flying activity begins at dusk and continues at night [116,117,118,119,120]. Moreover, under controlled conditions it was found that adult movement is strongly dependent on the illumination in their surroundings: it is almost instantly suppressed by light simulating daylight and activated almost immediately after darkness [121]. Accordingly, DBM adult activity during twilight and night is characterized by the phenomenon of positive phototaxis, which is inherent in both sexes. Therefore, capturing moths with light traps remains relevant for this species [122,123,124,125,126]. Until recently, the use of light traps for trapping the DBM and other microlepidoptera species was hindered by the bulkiness of their design and dependence on power sources [40,127]. However, thanks to the advent of LED technology, this situation has improved dramatically in recent years [126,128,129,130]. In comparison with gas-discharge and incandescent lamps, LEDs have a number of advantages: they are distinguished by a significantly longer service life, low energy consumption, higher luminous efficiency, adjustable color temperature, compactness, low heating, resistance to mechanical damage, and low maintenance costs [131,132,133,134]. And most important, LEDs are able to provide large insect captures in comparison with light sources of previous generations [135,136,137,138].

It has been shown that not only SSA but also LEDs can be used in traps both for monitoring and for suppressing DBM populations by mass-trapping adults, which significantly reduces insecticide treatments [139,140]. Given the relatively low selectivity of insect species for light attraction, low-intensity LEDs are currently considered the most promising for trapping harmful species. In fact, low-intensity LEDs exhibit a low level of danger to non-target entomofauna since traps equipped with these LEDs have the following characteristics: (1) they are easily placed in the habitats of target insect pests due to their compact size, and (2) their attractiveness is limited by rather short distances [141,142,143,144,145].

Taking into account the real possibility of achieving synergy, researchers are increasingly paying attention to combining different baits for insect trapping. In particular, it has been found that a mixture of semiochemicals of plant origin with DBM SSA not only can provide more effective monitoring but also guarantees an environmentally safe reduction in pest populations [146,147]. As for evaluating the potential of combining both SSA and LEDs in traps as a way to enhance the attractiveness of traps for DBM adults, no such work has yet been conducted. To date, only a few articles have compared DBM catches in traps equipped with either LEDs or SSA individually [140,148,149].

The purpose of this study is to conduct field trials of traps equipped with both SSA and UV LEDs to quantify effects of the interaction of these lures for capturing DBM adults.

## 2. Materials and Methods

### 2.1. Site of Experimentation

The trapping trials were conducted in the vicinity of St. Petersburg (northwestern Russia) at the experimental field of the Pushkin and Pavlovsk Laboratories of the N. I. Vavilov All-Russian Institute of Plant Genetic Resources (VIR) (PPL VIR) in the city of Pushkin (59°42′51″ N, 30°23′47″ E) (Figure 1) in 2022–2024.

The site on which samples of the VIR world collection of cabbage were grown in 2022–2024 had an area of at least 1500 m^2^. The area had previously been planted with squash, pattypan, and pumpkin. The collection material of the garden cabbage crops (white cabbage, red cabbage, leafy cabbage, Brussels sprouts, cauliflower, and broccoli), represented by different varieties and hybrids, was grown with a seedling method. As a rule, seedlings were planted in the ground in the first week of June. There were 20 plants per plot; row spacing was 70 cm; distance between plants within the plot was 60 cm. Three days before planting the seedlings in the ground, they were treated with Syngenta Aktara VDG (250 g/kg) to control cabbage flies, *Delia radicum* (L.) and flea beetles, and *Phyllotreta* spp. In 2022, 35 days after the seedlings were planted in the ground, a single treatment of the site was carried out with Syngenta Proklame VG (50 g/L) controlling a complex of lepidopteran pests. In 2023, 16 days after planting the seedlings in the ground, the site was treated once with Aktara VDG to control cabbage flies and flea beetles. In 2024, 16 days after planting the seedlings, the site was treated once with Keminova A/S Danadim Expert CE (400 g/L). Weeding, loosening, and fertilization with ammonium nitrate as well as row-by-row plant processing were also carried out during the growing season of the cabbage crop.

Figure 2 illustrates the meteorological conditions on the territory of the PPL VIR during the summer months of 2022–2024 based on data obtained at the meteorological station (see Figure 1).

### 2.2. Traps and Insect Baits

To carry out field trials to assess the attractiveness of a set of baits for DBM adults, we used Delta traps manufactured according to utility model patents RU 195732 U1 and RU 220753 U1 by the Innovation Center RIKSO LLC (St. Petersburg, Russia) (https://companium.ru/id/1037800125241-ic-rikso, accessed on 20 August 2025) (Figure 3). In the upper part of each trap made of transparent plastic, a removable cassette was installed, which contained a power supply (six AA 1.2 V batteries with a capacity of 2200 mAh each), a photosensor, a board with two low-intensity LEDs, control unit in the form of an Attiny 25 V microcontroller with a recorded LED control program and photoresistor connected to it, an illumination programming button, and an electric throttle. The removable cassette was easily installed in the trap and could be easily changed if necessary. Batteries were recharged using a solar panel mounted on one of the side faces of the trap. Considering that the DBM adult is actively attracted to UV radiation in the range of 365–400 nm [126,139,140], two low-intensity LEDs manufactured by Oumurui (Shenzhen, China) with a wavelength of 365–370 nm which radiated light in opposite directions from each other were installed in each trap. The rated power of each LED was 3 W, the electric current was 40 mA, and the estimated luminous flux in the UV range was 25–35 lm each. The LED power supply was controlled by an Attiny 25 V microcontroller. The program loaded into it provided control of the brightness of the LED glow by using the following: (1) pulse power supply with a frequency of 30 kHz; (2) software setting of the brightness of the LEDs; (3) switching on and off the LED at a preset (programmable) illumination level; (4) minimization of thermal energy loss of power sources; (5) disconnection of batteries to prevent their deep discharge; and (6) determination and display of battery charge level.

Previously, it was experimentally demonstrated that an LED power supply using high-frequency pulsed current contributes to a highly significant increase in DBM collection by light traps [150]. This result was achieved due to the psychophysical effects of flicker fusion at high frequency [151] and visible persistence [152], although the energy consumption when powering LEDs with pulsed and direct current was the same (Patent RU 220753 U1). The frequency of a pulsed LED power supply (and, accordingly, flicker) equal to 30 kHz was chosen based on the desire to minimize the size of the construction, the capabilities of microcontroller, and the time of switching on/off transients of LEDs, as well as an analysis of the literature on the effects of flickering lights on insect behavior [150,153].

A commercial product from Pherotrap LLC (Moscow, Russia) (https://pherotrap.ru/) was used as the source of SSA in the traps, namely, rubber dispensers impregnated with DBM SSA (a mixture of (Z)-11-hexadecenal and (Z)-11-hexadecenyl acetate in a ratio of 10:90 to which as a minor component (0.99% of the total composition) (Z)-11-hexadecene-1-ol was added). To trap the attracted insects, sticky cardboard manufactured by the same company was placed at the bottom of each trap. The SSA dispenser was placed in the center of the sticky cardboard. As a result, direct exposure of the light radiation from the LEDs on the SSA dispenser in the trap was excluded, which ensured the correct assessment of the effect of the interaction of SSA and UV radiation on insect attractiveness while using both baits in one trap.

The automatic switching on of the LEDs in the traps was adjusted to the actual observed illumination level during sunset, and the switching off was set for illumination during sunrise; these times were recorded annually before the start of trap trials in early June.

### 2.3. Experimental Design

In the trials, the following baits were tested in traps: (1) dispensers with SSA; (2) LEDs; (3) a combination of a dispenser with SSA and LEDs (SSA + LED); and (4) no bait in the trap (Control). The traps were placed on wooden stakes at a height of 60 cm above the ground and at a distance no closer than ten meters from each other and from the edge of the cabbage plot, in three randomized blocks (Figure 4). Traps were installed immediately after planting cabbage seedlings in the ground. The catches of DBM adults by traps were recorded until the second week of September over three seasons (2022–2024). Before the first DBM adult appeared in the traps, they were checked daily; subsequent counts were carried out at least twice weekly. During inspection of the traps, DBM adults caught were counted and removed from the sticky cardboard, and the sex of each individual was determined based on the external structure of their genitalia [154]. The sticky cardboard was replaced as it became dirty, and the SSA dispenser was changed once a month.

### 2.4. Statistical Analysis

It is well known that obtaining resulting data close to 0 in particular experimental variants (usually the control) is a problem when performing parametric statistical analysis [155]. Since even after transforming the data of DBM captures according to the recommendation [156], a significant portion of the distributions of captured insects did not meet the requirements for correct implementation of ANOVA (normality of distributions according to the Shapiro–Wilk test and uniform variances according to the Levene and Cochran criteria), non-parametric Kruskal–Wallis and Mood’s Median tests were used for statistical analysis. After proving the validity of the bait effect, the significance of differences between the averages was evaluated according to the Wilcoxon–Mann–Whitney pairwise test, post hoc Bonferroni-corrected for multiple comparisons at 5% significant level (α = 0.05). To assess the effects of the interaction of the attractive properties of UV LED and SSA, the Adjusted Rank Transform Test was used [157], which is recommended for verifying the validity of interactions between factors with non-normal distributions [158,159,160,161].

To determine if the ranking of DBM adult capture distributions across various treatments consistently follows a monotonic pattern, we employed both Friedman and Page [162] tests (the latter was assessed to prove the significance of the linearity effect in ranking). To visualize the variation in DBM captures using traps with different treatments during a weekly period, diagrams were made both over variations for each year and over all three years together (in the form of a box-and-whisker plot). The quantitative assessment of increased DBM trapping resulting from combining two baits (SSA and LED) is characterized by a multiplicity of increases in moth collection due to retrofitting with second bait (LED) of traps previously equipped with SSA, as this bait is used practically constantly for trapping DBM adults, e.g., [163,164]; therefore, only such an indicator allows a comparison of the results obtained with the literature data. To perform all the abovementioned work, MS Excel 2024, Statgraphics Centurion 19, and Tibco Statistica 14 were used.

## 3. Results

### 3.1. Statistics of DBM Adult Captures Using Delta Traps with Different Baits

The results of capturing DBM adults at the experimental site of the VIR cabbage collection using traps with four treatments (Figure 5) are presented in Table 1. During a period of three years (2022–2024), a total of 14,655 DBM adults were captured, with the vast majority being males (98.5%). The SSA + LED traps captured almost three times more DBMs than LED alone and almost 13 times more than SSA alone (Table 1).


The results of statistical analysis of data on the catches of DBM adults using traps equipped with different baits for each year and a total of three years’ worth of trials are presented in Table 2. These data indicate a statistically significant difference (Kruskal–Wallis and Mood’s Median tests) in average capture rates between males and females using different baits. It is important to note that there was a highly significant synergistic interaction between the SSA and LED baits in terms of their attraction to DBM males, as determined by the Adjusted Rank Transformation Test (*F* = 8.56, *p* < 0.001). As for the females, the SSA did not appear to be attractive, as expected, while the LEDs were attractive but approximately 55 times less than for males on average over 3 years of trials. Consequently, the interaction effect of SSA and LED on females was statistically insignificant (*F* = 1.39, ns) in contrast to males. However, if the catches of males and females are considered in total, i.e., without taking into account the sex of the captured adults, then the synergy effect of SSA and LED attractions can be proven at a high level of significance (*F* = 7.27, *p* < 0.01). This is not surprising, since males are attracted to traps many times more often than females: in three years, the proportion of females captured in traps with different treatments was estimated at only about 1.5% (Table 1).

### 3.2. Monotonic Relationship Among the Distributions of DBM Catches with Different Treatments in Traps

To more accurately characterize the variation in DBM adult trapping trials, it is important to assess whether the order of DBM adult capture distributions across different treatments remains consistent over time. For this purpose, we used the averages across three randomized blocks of DBM adult captures during the 2022–2024 growing seasons (Figure 6) for analysis using the Friedman test. This test clearly validated the robust statistical differences between treatment rankings based on DBM captures: *χ*^2^ values were 30.28 in 2022, 30.51 in 2023, and 31.61 in 2024 (*df* = 3, *p* < 0.000001 in all cases). Furthermore, we used the Page test [162] to verify the consistent monotonicity of DBM adult catch distributions across the four treatments, i.e., effect of linearity in rankings. First, according to the gradations of the distributions of trapping treatments by quartiles, non-outlier ranges, and outliers of captured moths (Figure 7), each treatment was assigned a rank from 1 to 4: 1 for Control, 2 for SSA, 3 for LED, and 4 for SSA + LED. In the result, the Page test confirmed a highly significant linear component in the ordering of the four treatments over time: *L* = 375 in 2022, *L* = 379 in 2023, and *L* = 378 in 2024 (*n* = 4, *m* = 13, and *p* < 0.001 in each analysis).

### 3.3. The Increase in the Number of DBM Adults Collected by Traps Due to the Interaction of Bait Attractiveness

As mentioned above, the increase in insect trapping as a result of combining two baits in a trap was usually characterized in publications by a multiplicity of increases in insect captures due to the retrofitting with additional bait of traps, which were already equipped with the main bait conventionally used for trapping target species. The main bait normally used to catch DBM adults is SSA [60,82,83,84,85,86,87]. Therefore, in order to compare our results with the literature data, we estimated the effect of retrofitting SSA-based traps with LED as the second bait on an increase in DBM adult captures (Table 3). If we use data on whole-moth captures from traps equipped with SSA, LED, and SSA + LED (Table 1 and Table 2) to quantify the synergistic effect, the multiplicity of adult collection increases ranges from 11.2 to 19.2 when retrofitting SSA-based traps with LED bait. On average, this gives a 15.1-fold increase (Table 3).

### 3.4. Correlation Analysis of DBM Adult Captures Using Traps with Different Baits

The findings of the correlational analysis (Table 4) primarily suggest that the traps equipped with SSA and LED exhibit distinct patterns in the population dynamics of DBM adults. Secondly, the data presented in Table 4 underscore the consistency in the manifestation of statistically significant correlations between the numbers of captured DBM adults in traps with SSA and LED on the one hand and in SSA + LED-equipped traps on the other hand. This observation implies that employing traps equipped with SSA + LED can provide a comprehensive overview of pest population dynamics, which differs from that obtained by using SSA or LED traps individually (Figure 6), making it a valuable tool for monitoring purposes.

## 4. Discussion

Insect traps, which come in a variety of designs, have long been widely used in plant protection practices, both as a monitoring tool and as a means of controlling crop pests [165]. Traps typically use various methods of attraction to manipulate the behavior of insects by influencing their sensory systems [166,167]. The use of these traps to manipulate pests’ behavior fully aligns with the principle of using approaches that prioritize the safety of the environment [168].

In accordance with the specifics of the effect on the sensory organs of insects, the means of attraction are usually divided into attractants of a chemical (semiochemicals) or physical (semiophysicals) nature [169]. Among semiochemicals, a diverse range of compounds have been identified that control a wide array of insect behaviors, with SSA compounds, specifically providing intraspecific communication, most frequently employed in plant protection practices [170,171]. Semiophysicals, on the other hand, encompass a variety of distinct signals, such as sound impulses and vibrations [169], although light radiation is predominantly employed for the purposes of pest management [172].

The joint functioning within traps of attraction tools with diverse modalities holds promise, offering two key advantages: (1) a substantial enhancement in the dependability of parent–offspring regression relationships as a predictive framework for monitoring models when low population occurs, and (2) a substantial reduction in labor expenses for mass fishing operations due to a diminished requirement for traps to achieve the desired level of pest reduction. Consequently, the integration of attractants with various modalities within traps, resulting in a multimodal influence on insect behavior, offers the potential for improved trap efficacy both in monitoring tasks and in pest management efforts [172]. Since the mid-20th century, scientists have been endeavoring to enhance the effectiveness of traps by integrating light signals with specific odorous lures. This approach has proven successful in capturing males of various moth species, such as cabbage looper, *Trichoplusia ni* (Hbn.) [173,174], tobacco hawk moth, *Manduca sexta* (L.) [175,176,177], and tobacco budworm, *Chloridea virescens* (F.) [178], when virgin females were placed into a specially designed chamber within the light trap.

The data presented in this paper strongly suggests that when SSA and LED signals are combined, they can produce a synergistic enhancement in the capture of DBM males using Delta traps. As a result of retrofitting SSA-based traps with LEDs as an additional lure, catches of DBM adults showed an average 15-fold increase (Table 3).

The findings presented in this study are of interest for the following reasons: (i) their contribution to our knowledge of ranges in interactions between different types of insect lures, (ii) their potential applications in improving DBM monitoring, and (iii) the possible enhancement of the efficiency of mass-trapping techniques to control the pest.

### 4.1. Interactions of Baits with Various Insect Attraction Modalities

There is a large amount of evidence in the literature for the increased efficiency of using a combination of various attractants—semiochemicals (pheromones, allomones, kairomones, and synomones) and semiophysicals (primarily light radiation) to trap harmful insects, which achieves an increase in attraction [169,172,179]. For example, when traps were equipped with a combined bait of SSA and LEDs, it was possible to increase the catch of sweet potato weevil males, *Cylas formicarius* (Fabr.), almost five-fold [164]. In addition, traps equipped with UV-emitting lamps and SSA attracted six times more cabbage looper (*Trichoplusia ni*) males than traps with UV lamps alone [163], and the combination in traps of UV LED with semiochemicals resulted in a 2–12-fold increase in catches of adults of four harmful Lepidoptera species, including orchard codling moth, *Cydia pomonella* (L.), oriental fruit moth, *Grapholita molesta* (Busck), oblique banded leafroller, *Choristoneura rosaceana* (Har.), and eye-spotted budmoth, *Spilonota ocellana* (Den. et Schiff.). Interestingly, when low-intensity LEDs were used alone for trapping, no insects were attracted [145]. The increase in attractiveness when combining semiochemicals and semiophysical attractants has also been shown to be rather synergistic for some other insect species, including red flour beetle, *Tribolium castaneum* (Herbst) [180], cigarette beetle, *Lasioderma serricorne* (F.) [181], brown marmorated stink bug, *Halyomorpha halys* (Stål) [182,183], western flower thrips, *Frankliniella occidentalis* Perg. [184], midnight woodwasp, *Sirex noctilio* F. [185], and several species of sink flies from the Psychodidae family [186,187]. In some cases, there were other effects of the interaction between light and chemical signals, including additive, non-additive, and even antagonistic ones [188,189,190,191]. Although the latter are rarely mentioned in the literature, a comparative analysis of published information on the scale of increase in insect trapping as a result of combining semiochemicals and light radiation shows that the highest recorded synergy effect is a nine-fold increase in captures of the tomato leafminer, *Tuta absoluta* (Meyrick) [192]. However, the data presented in this article suggests that the current record now belongs to the DBM, with a 15-fold increase in catches after SSA traps were retrofitted with LEDs.

### 4.2. Prospects for Combining SSA and LED in Traps for DBM Monitoring

In the field of pest management, insect traps have long been used primarily to monitor harmful insects [193]. Until now, light traps continue to be widely used for ecological and faunal research, as well as for studying insect distribution patterns and phenology, providing long-term data on population dynamics across various regions worldwide [194,195,196,197,198]. However, in the context of routine phytosanitary surveillance for insect pests in agriculture, light traps have been superseded by pheromone traps since the 1970s and 1980s due to their inferior specificity, higher cost, and limited mobility [199,200,201]. At present, owing to the exceptional properties of LED technology, there is a trend towards an increasing use of light traps for monitoring different agricultural pests. This trend is gaining momentum due to the efficiency of LED-based solutions in this field [137,138,202].

It is well known that pheromone-based products exhibit a wide range of unique properties, including selectivity of action, high efficacy, low toxicity, high volatility, and short persistence [203,204,205,206,207]. However, the degree of attractiveness of these products to insects varies significantly depending on various factors, such as weather or climatic conditions, the specific composition of the pheromonal attractant, trap design and manufacturing techniques, characteristics of insect reproductive behavior, and the state of the target pest population [111,207,208,209,210,211,212,213,214]. Similarly, the attractiveness of light-based stimuli for insects also varies significantly depending on many factors and especially on the level of natural illumination [209,215,216,217,218,219,220]. Also, since pheromone and light traps are equipped with attraction devices with fundamentally different modalities, their attractiveness varies in a completely different way. So, it has been repeatedly recommended to use both types of traps together to achieve a more accurate prediction of the population dynamics of the number of harmful insect species [209,221,222,223,224,225]. It is likely that similar results can be achieved by combining light and pheromone baits within a single trap, especially because in the case of synergistic or even additive effects of the lures, in addition to increased reliability of monitoring, sometimes one can also expect an increase in the negative impact of the traps on the density of pest populations [145,164,180,181,182,183,184]. As for the DBM, given the exceptional additive interaction of LED and SSA baits in terms of attractiveness to the adults of this species, the advantages of monitoring the pest using traps equipped with a combination of these baits seem quite evident. Furthermore, as the results of the correlation analysis (Table 4) demonstrate a high level of consistency in statistically significant associations between the numbers of captured DBM adults in traps with SSA or LED on the one hand and SSA + LED on the other hand, the utilization of traps equipped with SSA + LED allows us to anticipate obtaining a more comprehensive understanding of the population dynamics of the pest, which is displayed differently by traps equipped with either SSA or LEDs.

### 4.3. The Potential of Combining SSA and LED for Mass Trapping of DBM Adults

In addition to monitoring, SSAs are used as a means to control the number of harmful insect species, which is achieved through the use of various technologies such as mass trapping, attract-and-kill, mating disruption, and push–pull. Of these, the first two techniques, which are similar in principle, have certain advantages over the latter two, as they are more cost-effective, requiring much smaller amounts of pheromone-based substances [111,171,204]. Although mating disruption technology occupies a leading position throughout the world in terms of its application due to its high efficacy and adaptability [207,226,227,228], mass-trapping technology also holds promising prospects [229]. For instance, while attract-and-kill systems that utilize gel drops require insects to come into direct contact with the odor source [230], traps designed for mass trapping typically do not necessitate such behavior of the target object [204]. Although pheromones that attract individuals of both sexes of a harmful species have a stronger negative impact on population dynamics, SSA attracting only males can also suppress pest reproduction due to the fact that male removal prevents mating with females, thus preventing the development of offspring that damage plants. Despite some challenges associated with the implementation of mass-trapping techniques using SSA [172], a significant number of successful examples have been documented in the literature, both in standalone use and in combination with other control approaches [111,204].

The literature contains a wealth of data that suggests the efficiency of employing DBM SSA as a tool for controlling the pest through the implementation of mass-trapping techniques. This approach effectively depletes the male population, leading to delays, reductions, and avoidance of fertilization in females, ultimately resulting in a decrease in the density of larvae that feed on plants, which, in turn, reduces plant damage and promotes crop growth [100,101,102,103,104,105,106].

One of the first attempts to assess the feasibility of using pheromone traps as a method for large-scale DBM trapping was the testing of SSA-equipped Pherocon^®^ 1CP traps in Canada in the late 1970s [69]. Experiments conducted in India in the late 1980s demonstrated that sticky Delta traps equipped with SSA, installed in experimental cabbage fields (six traps per field), caught between 11 and 13 thousand DBM males per season, resulting in a significant reduction of plant damage caused by the pest and, consequently, a substantial increase in cabbage yield [100]. Subsequently, Chinese researchers found that the use of three SSA-based traps per 180 m^2^ greenhouse ensured the capture of more than 1800 DBM adults per season, allowing for the achievement of 70% efficiency in suppressing the DBM population, albeit under conditions of low pest egg density (one to three eggs per plant) [101]. In a mountainous region of China, traps containing SSA were shown to catch between 11 and 70 moths daily, significantly reducing the population of the DBM and preventing the need for insecticide treatments to protect cabbage [103]. Experiments conducted in Vietnam evaluated the efficiency of three types of traps and two SSA combinations, demonstrating the potential for mass trapping of DBM adults in the tropics. In these experiments, the maximum catches of male moths (up to 170 or more individuals per day) were obtained using traps in the form of plastic basins filled with water [102]. The efficiency of mass trapping as a standalone method to control the DBM was assessed over two seasons in India on cabbage, also using SSA-based traps in the form of basins filled with water. In the first season, 675 moths were captured by 21 traps in the experimental cabbage field (i.e., an average of 32.14 individuals per trap), and in the second season, 13,935 moths were captured by 24 traps (an average of 580.60 individuals per trap). As a result, the maximum reduction in the number of DBM larvae and pupae on the experimental plants compared to the control was observed in plots where moth trapping was used: in the first season, this resulted in a decrease of 73.96% in the number of larvae and pupae compared to the control, while in plots where conventional farming methods were used (five insecticide treatments per season), the decrease was only 46.11%. In the second season, mass moth trapping resulted in a 51.42% decrease in larvae and pupae compared to the control, whereas using conventional farming methods resulted in a population decrease of only 39.04%. The largest percentage reduction in the DBM population compared to the control was observed during mass trapping as a standalone method. This was largely due to the significant increase in the population of the DBM parasitoid *Cotesia vestalis* (Hal.) as opposed to plots where conventional farming methods were used. Based on calculations using the data obtained, it was recommended that traps should be placed in fields at a density of 40 per acre (98.8 per one ha) for mass trapping as a standalone technique [104]. As part of the standardization of using SSA-based traps for mass trapping of DBMs in India, it was shown that the maximum number of DBM adults (15,394 moths) was trapped in a cabbage field with a density of 24 traps per acre (i.e., 59.3 per ha). Since SSA traps actively attract DBM adults for 60 days and during this time farmers typically apply 8–10 insecticide treatments to cabbage, each trap installed on the field prevents 4–5 chemical insecticide applications per acre throughout the year [105]. Experiments conducted in Costa Rica with water-filled SSA-based traps placed at a density of 50 traps per one ha confirmed their high efficacy in controlling the DBM compared to insecticide treatments, as they provided a profit of USD 1240 per one ha [231]. Another publication presenting the results of studies conducted not only in Costa Rica but also in Nicaragua convincingly confirms the high efficacy of using SSA-based traps not only for monitoring but also for mass trapping of DBM males, which recommends placing traps with SSA at a density of 60 per ha in cabbage fields [106]. Therefore, the test results for SSA-based traps conducted under a diverse range environments indicate that the mass trapping of DBM males on cabbage fields at a density between 50 and about 100 traps per ha has the potential to reduce DBM populations to a level comparable to or even surpassing that achieved through the use of chemical insecticides.

The use of light traps as an alternative to chemical control of harmful insects predates the development of pheromone traps by several decades, dating back to the 1920–1930s, and often, their use proved to be highly effective [232,233]. Despite some advantages of light traps over SSA-equipped traps, such as their ability to attract a wider range of species, their trapping of not only males but also females, their longer lifespan, and other benefits, they were eventually superseded by pheromone traps in the 1970s due to the superior specificity, affordability, and portability of SSAs-based traps [234]. However, recent advancements in LED technology have sparked renewed interest in the utilization of light traps for pest management [125,130,137,234].

Experiments conducted in Indonesia using LED traps for mass trapping demonstrated their effectiveness in controlling not only the DBM but also other pests such as cabbage webworm, *Crocidolomia pavonana* (F.), and tobacco cutworm, *Spodoptera litura* (F.). The experiments were carried out using UV-emitting LED (365–385 nm) traps, which were designed as water-filled containers. Experimental cabbage plots were divided in half: one group of plots had light traps installed, and the other group did not. Plants in the first group of plots (with light traps installed) were periodically inspected for pest presence, and if the density of DBM larvae or *C. pavonana* eggs exceeded a harm threshold, the plants were treated with insecticide. Plants in the second (control) group of plots (without light traps) were sprayed with insecticide twice weekly invariably. DBM, *C. binotalis*, and *S. litura* adults were detected in traps installed on cabbage plots as early as four days after planting the cabbage seedlings. The number of DBM adults averaged 25 individuals per trap at this time, and 46–50 days after planting it exceeded 50 individuals per trap. The results obtained indicate that plants on plots with light traps needed only two insecticide sprays throughout the entire six-month test period, since the density of harmful species’ preimaginal stages on these plants was maintained below the harm thresholds. This reduced the number of insecticide treatments by 81.82% compared to the control, and the yield of cabbage on these plots was equivalent to the yield in the control. Therefore, the use of LED traps to protect cabbage from lepidopteran pests has proven to be economically profitable [139]. Further studies carried out in Indonesia aimed to compare the effectiveness of LED-equipped traps, DBM SSA, and SSA of *C. pavonana* in protecting cabbage from lepidopteran pests. The results showed that LED traps started attracting moth adults earlier than traps with either type of SSA and caught them more effectively. As a result, plots protected with LED traps had lower densities of harmful larvae and higher yields (1.75 kg per plant) than plots protected with SSA traps (1.18 kg per plant with DBM SSA and 0.92 kg per plant with SSA of *C. pavonana*) [140]. In the articles cited above, there is no information regarding the sex of moths caught in light traps. However, even if light traps in experiments conducted in Indonesia caught almost exclusively males, as in trials conducted in St. Petersburg (see Table 1), their negative impact on the DBM population was significantly greater than that of traps equipped with SSA, which was confirmed by comparing pest densities on plants and crop yields.

Based on the analysis of published materials, it can be concluded that SSA-based traps designed to capture exclusively males can significantly reduce the DBM population. With regard to LED-based trapping systems for DBM mass trapping, these appear to hold even greater potential in controlling pest harmful activity. However, it is clear that LED-based traps are more expensive than traps with SSA. Despite this, considering the great increase in DBM adult captures achieved by retrofitting SSA-based traps with LEDs (estimated at a 15-fold increase), it seems likely that a mass-trapping approach utilizing SSA + LED traps could effectively suppress DBM populations at a relatively lower cost. This is because the number of SSA + LED-based traps per hectare of cabbage cultivation required would be significantly lower than that needed when using SSA alone to effectively control the pest population.

It should be noted that the level of SSA and LED interaction, estimated by a 15-fold increase in trap catchability, was obtained under conditions of high latitudes. In these areas, highly varying natural illumination at night during the season prevents the LEDs from achieving maximum attraction of night-flying insects. Hence, at lower latitudes with their consistently low levels of natural illumination at night, the level of SSA and LED interaction may be even stronger. However, it is evident that through the use of SSA + LED traps, an effective reduction in DBM populations can be achieved when fewer traps are deployed per hectare compared to traditional SSA traps (which are deployed at 50–100 per hectare). This will further enhance the profitability of mass-trapping efforts for DMB management.

## 5. Conclusions

Trials of traps equipped with four treatments (SSA, UV LED, SSA + UV LED, Control) and installed in cabbage fields were conducted in the vicinity of St. Petersburg, Russia, for three years (2022–2024). The trials were successful in demonstrating a very high level of synergy between SSA and UV LEDs in attracting DBM adults, resulting in a 15-fold increase in DBM adult (mostly represented by males) catches. Analysis of published data indicates that this effect is likely the most substantial of those reported in the literature. In addition to its monitoring value, the combination of SSA + LED, which is highly attractive to DBM adults, makes it promising for use in mass-trapping technology.

## 6. The Patents Employed in the Production of Traps

Miltsen, A.A.; Grushevaya, I.V.; Kononchuk, I.V.; Malysh, Y.M.; Tokarev, Y.S.; Frolov, A.N. Light trap for insect monitoring. Utility model patent RU 195732 U1, 2020-02-04.

Zakharova, Y.A.; Miltsen, A.A.; Frolov, A.N. LED insect trap with pulse power supply. Utility model patent RU 220753 U1, 2023-10-02.

## Figures and Tables

**Figure 1 insects-16-00881-f001:**
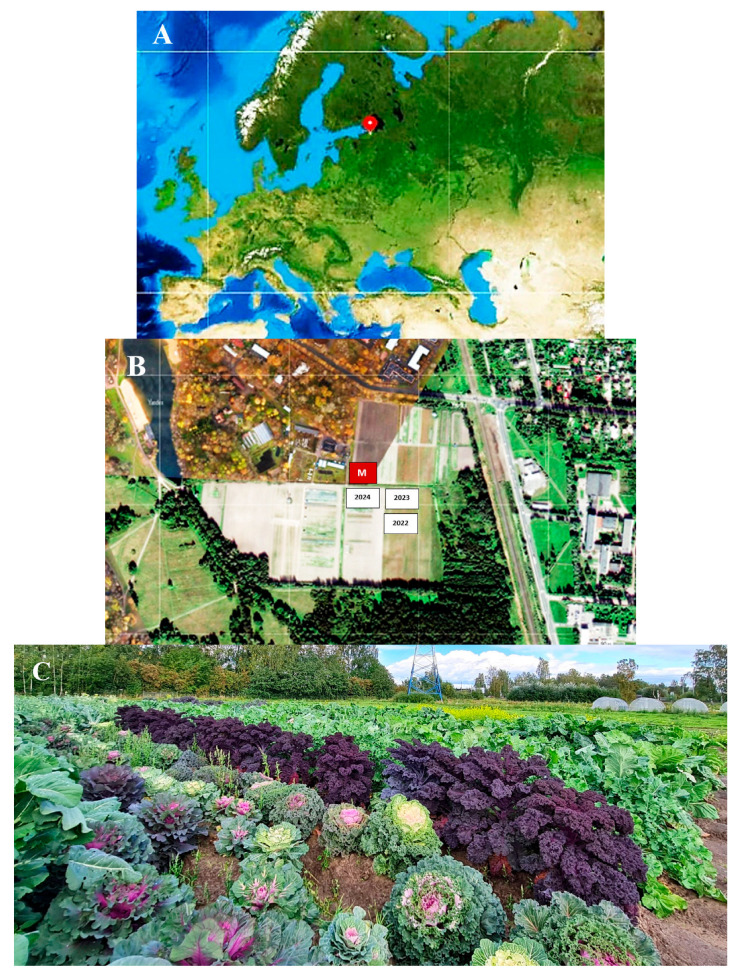
Area under experimental study at PPL VIR (Pushkin, near St. Petersburg, northwestern Russia). (**A**) Geographical location of the place of work (the vicinity of St. Petersburg). (**B**) Google map of the part of the city of Pushkin where the cabbage collection was grown and where field trials of traps equipped with various baits for catching DBM adults were carried out. M—meteorological station; 2022, 2023, and 2024—plots occupied by cabbage in 2022, 2023, and 2024. (**C**) General view of the site where the VIR world collection of cabbage was grown and where the traps for catching DBM adults were placed.

**Figure 2 insects-16-00881-f002:**
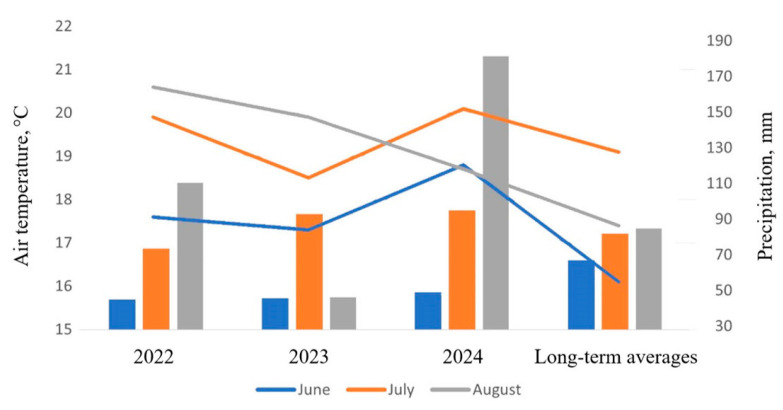
Meteorological conditions in the summer months of 2022–2024 in the territory of the PPL VIR, St. Petersburg. **Lines**: average monthly air temperatures, °C. **Columns**: monthly precipitation, mm.

**Figure 3 insects-16-00881-f003:**
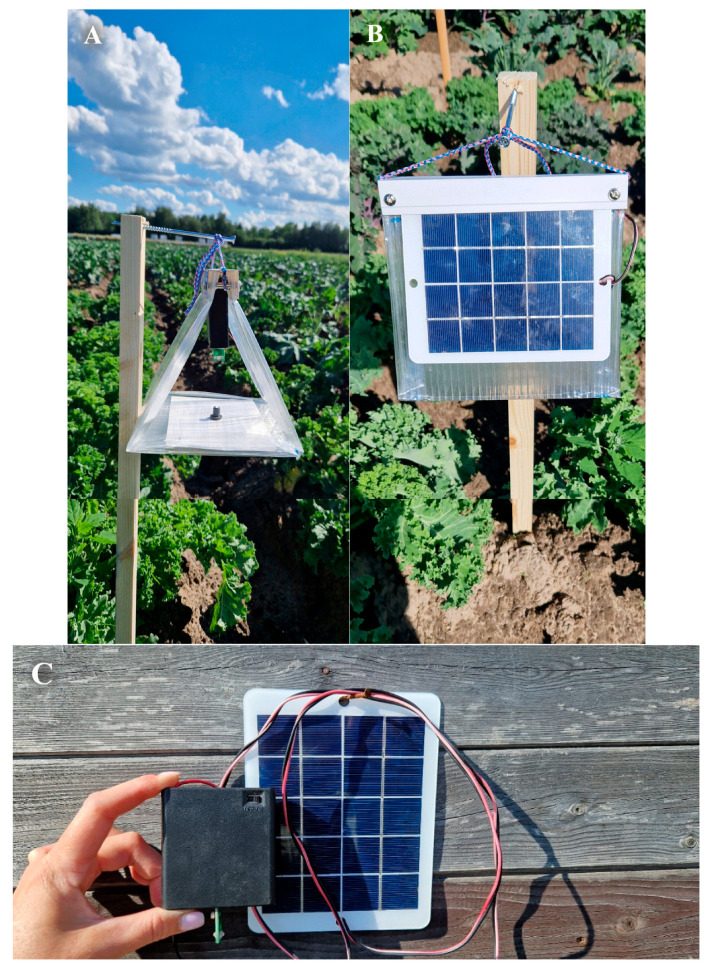
The Delta trap used for catching DBM adults. (**A**) SSA + LED experiment option, full-face view. A cassette with LEDs is installed at the top of the trap, and an SSA dispenser is installed at the bottom. (**B**) The same, profile view. A solar battery is fixed on the side of the trap. (**C**) The LED cassette and solar battery are shown separately from the trap body.

**Figure 4 insects-16-00881-f004:**
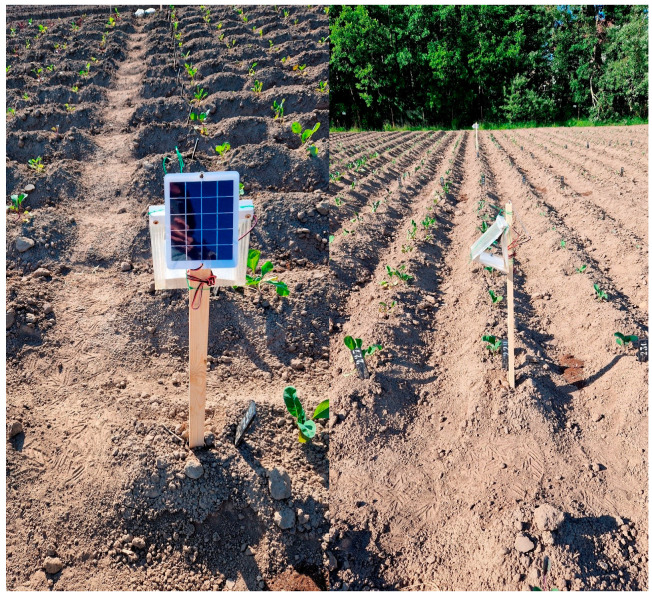
Delta traps placed on wooden stakes in a cabbage plot at no closer than ten meters from each other.

**Figure 5 insects-16-00881-f005:**
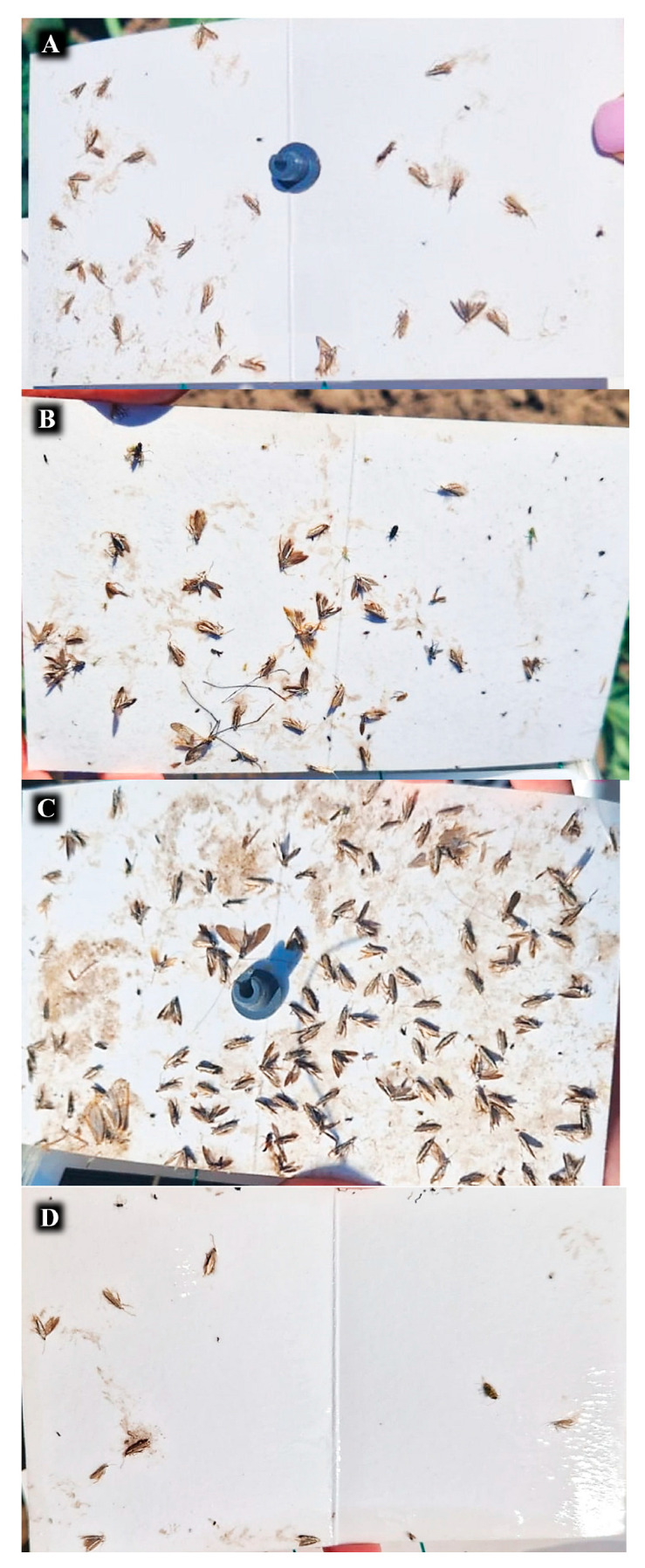
Randomly selected photos of sticky cardboard containing insects captured, which were in traps with different treatments for the same three days. (**A**) Dispenser with SSA. (**B**) LED. (**C**) SSA + LED. (**D**) Without lure (Control).

**Figure 6 insects-16-00881-f006:**
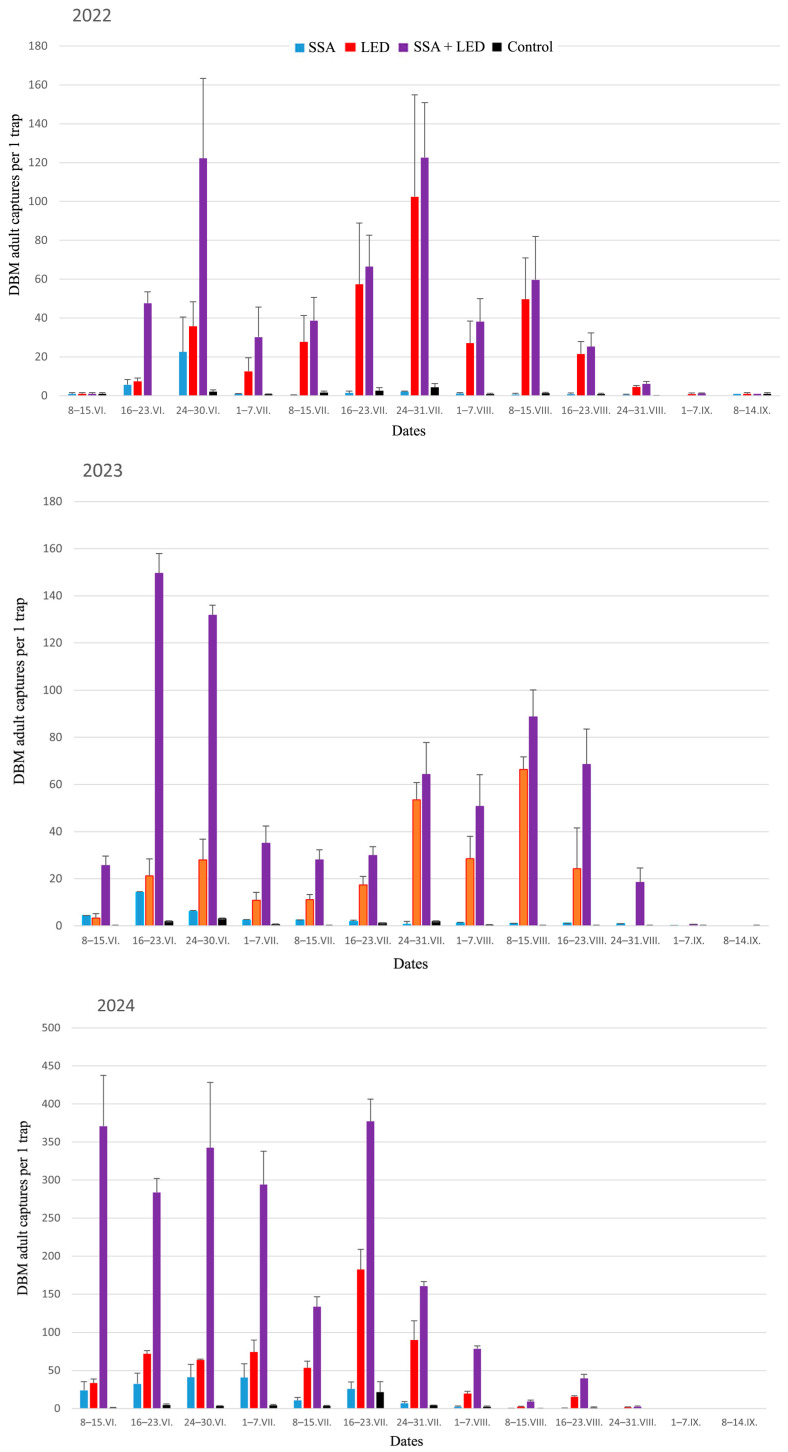
Dynamics of DBM adult (males + females) captures (*x* ± *SE*) with the four different treatments over a weekly period from 8 June to 14 September each year in 2022–2024. SSA = synthetic sex attractant, LED = light-emitting diode, and Control is traps equipped with neither of these.

**Figure 7 insects-16-00881-f007:**
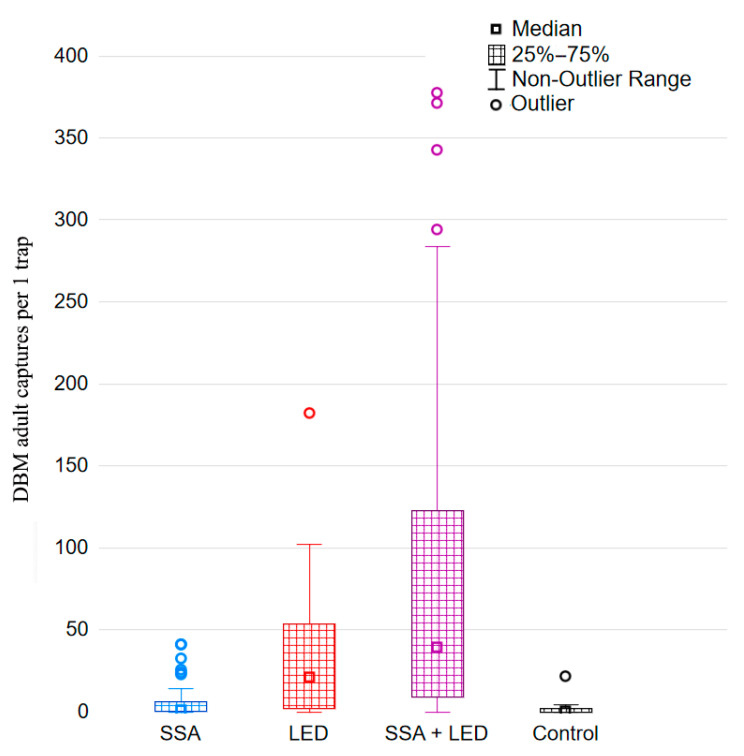
Box-and-whisker plot of DBM adult (males + females) captures with the four different treatments over a weekly period in each of the three years (2022–2024). SSA = synthetic sex attractant, LED = light-emitting diode, and Control is traps equipped with neither of these.

**Table 1 insects-16-00881-t001:** Captures of DBM adults using Delta traps equipped with four combinations of lures (PPL VIR, Pushkin, 2022–2024).

Bait	Sex	Captures					
		2022	2023	2024	In Total for 3 Years	Females, %	Ratio of Captures, %
SSA	males	113	109	553	775	0.13	5.3
	females	0	1	0	1		
	sum	113	110	553	776		
LED	males	964	798	1769	3531	2.57	24.7
	females	27	8	58	93		
	sum	991	806	1827	3624		
SSA + LED	males	1629	2089	6214	9932	1.17	68.6
	females	41	8	69	118		
	sum	1670	2097	6283	10,050		
Control	males	38	26	132	196	4.39	1.4
	females	4	0	5	9		
	sum	42	26	137	205		
Total	males	2744	3022	8668	14,434	1.51	100
	females	72	17	132	221		
	sum	2816	3039	8800	14,655		

**Table 3 insects-16-00881-t003:** The multiplicity of the increase in DBM adult captures after retrofitting SSA-based traps with LEDs as the second bait.

Year	The Multiplicity of Catch Growth Calculated by Sex	
	Males	Males + Females
2022	14.4	14.8
2023	19.2	19.1
2024	11.2	11.4
Mean	14.9	15.1

**Table 4 insects-16-00881-t004:** Matrix of Pearson correlation coefficients for the results of capturing DBM adults using Delta traps with different treatments in 2022–2024.

Treatment	LED	SSA + LED	Control
2022			
SSA	−0.135 (*p* = 0.59)	**0.546** ^1^ (*p* = 0.02)	0.036 (*p* = 0.88)
LED		**0.654** (*p* = 0.003)	0.814 (*p* = 0.00004)
SSA + LED			0.558 (*p* = 0.02)
2023			
SSA	0.021 (*p* = 0.93)	**0.656** (*p* = 0.002)	0.330 (*p* = 0.17)
LED		**0.541** (*p* = 0.02)	0.352 (*p* = 0.14)
SSA + LED			0.758 (*p* = 0.0002)
2024			
SSA	0.490 (*p* = 0.02)	**0.745** (*p* = 0.0001)	0.246 (*p* = 0.27)
LED		**0.590** (*p* = 0.004)	0.686 (*p* = 0.0004)
SSA + LED			0.267 (*p* = 0.23)

^1^ The cells with statistically significant values of the Pearson’s correlation coefficients obtained during all three years of trials are marked in boldface.

**Table 2 insects-16-00881-t002:** Comparison of DBM adult captures (per seven days) using Delta traps baited with four combinations of lures (with or without SSA and UV LED).

Year	SSA	LED	Mean (±SE) Catch per 1 Trap per 7 Days		
			Males	Females	Males + Females
2022	Yes	Yes	48.11 ± 14.14 a ^1^	1.21 ± 0.11 a	49.32 ± 14.22 a
		No	3.34 ± 1.78 bc	0 b	3.34 ± 1.78 bc
	No	Yes	28.47 ± 14.14 ab	0.80 ± 0.14 ab	29.27 ± 12.86 ab
		No	1.12 ± 0.16 c	0.12 ± 0.12 b	1.24 ± 0.23 c
KWT ^2^, *df* = 3			*χ*^2^ = 8.95 *p* < 0.05	*χ*^2^ = 10.17 *p* < 0.05	*χ*^2^ = 8.95 *p* < 0.05
MMT, *df* = 3			*χ*^2^ = 12.00 *p* < 0.01	*χ*^2^ = 12.00 *p* < 0.01	*χ*^2^ = 12.00 *p* < 0.01
2023	Yes	Yes	54.77 ± 4.42 a	0.21 ± 0.05 a	54.98 ± 4.43 a
		No	2.86 ± 0.74 bc	0.03 ± 0.03 b	2.88 ± 0.76 bc
	No	Yes	20.92 ± 1.32 ab	0.21 ± 0.07 a	21.13 ± 1.34 ab
		No	0.68 ± 0.23 c	0 b	0.68 ± 0.23 c
KWT, *df* = 3			*χ*^2^ = 10.38 *p* < 0.05	*χ*^2^ = 8.68 *p* < 0.05	*χ*^2^ = 10.38 *p* < 0.05
MMT, *df* = 3			*χ*^2^ = 12.00 *p* < 0.01	*χ*^2^ = 9.26 *p* < 0.05	*χ*^2^ = 12.00 *p* < 0.01
2024	Yes	Yes	172.61 ± 15.65 a	1.92 ± 0.29 a	174.53 ± 15.37 a
		No	15.36 ± 4.73 bc	0 c	15.36 ± 4.72 bc
	No	Yes	49.14 ± 3.12 ab	1.61 ± 0.25 ab	50.75 ± 2.87 ab
		No	3.67 ± 1.25 c	0.14 ± 0.07 bc	3.80 ± 1.31 c
KWT, *df* = 3			*χ*^2^ = 10.38 *p* < 0.05	*χ*^2^ = 9.74 *p* < 0.05	*χ*^2^ = 10.39 *p <* 0.05
MMT, *df* = 3			*χ*^2^ = 12.00 *p <* 0.01	*χ*^2^ = 12.00 *p* < 0.01	*χ*^2^ = 12.00 *p* < 0.01
2022–2024	Yes	Yes	91.83 ± 21.15 a	1.11 ± 0.26 a	92.68 ± 21.20 a
		No	7.18 ± 2.52 c	0.01 ± 0.01 b	7.19 ± 2.52 c
	No	Yes	32.84 ± 5.79 b	0.87 ± 0.22 a	33.57 ± 5.80 b
		No	1.82 ± 0.59 c	0.08 ± 0.04 b	1.90 ± 0.62 c
KWT, *df* = 3			*χ*^2^ = 26.80 *p* < 0.00001	*χ*^2^ = 11.05 *p* = 0.01	*χ*^2^ = 26.89 *p* < 0.00001
MMT, *df* = 3			*χ*^2^ = 23.56 *p* < 0.0001	*χ*^2^ = 11.31 *p* = 0.01	*χ*^2^ = 23.56 *p* < 0.0001
ARTT, *df* = 1, 24			*F* = 8.56 *p* < 0.01	*F* = 1.39 *ns*	*F* = 7.27 *p* = 0.01

^1^ Values are provided with the same letters when do not differ at *p* < 0.05 according to Wilcoxon–Mann–Whitney pairwise test, post hoc Bonferroni-corrected for multiple comparisons; ^2^ KWT—Kruskal–Wallis test; MMT—Mood’s Median test; ARTT—Adjusted Rank Transform Test.

## Data Availability

The original contributions presented in the study are included in the article, further inquiries can be directed to the corresponding author.

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
