# Peer review of "Highly Improved Captures of the Diamondback Moth, Plutella xylostella, Using Bimodal Traps"

_insects, 2025, doi:10.3390/insects16090881_

Round 1

Reviewer 1 Report

Comments and Suggestions for Authors

The proposed methodology is considered adequate, well designed, and consistent with the objectives sought, as the results obtained and their statistical consistency indicate the methodology is correct. To verify repeatability of the results, the sampling sites can be changed and replicated in other areas of the world, taking advantage of the insect's widespread distribution and thus verifying the method's effectiveness. However, for the purposes of this study, the work is considered to meet its objectives.

Author Response

Author's Reply to the Review Report (Reviewer 1)

Comments and Suggestions for Authors

The proposed methodology is considered adequate, well designed, and consistent with the objectives sought, as the results obtained and their statistical consistency indicate the methodology is correct. To verify repeatability of the results, the sampling sites can be changed and replicated in other areas of the world, taking advantage of the insect's widespread distribution and thus verifying the method's effectiveness. However, for the purposes of this study, the work is considered to meet its objectives.

Response: Dear Reviewer 1,

Thank you very much for your kind feedback on our article. Of course, the authors of the article agree with you that the results presented in the article need to be validated, and the validation must be carried out in different ecological and geographical locations of the planet, and especially in low latitudes. We are very interested in this since the conditions in high latitudes where we conducted our research were not favorable to ensure a high level of attractiveness for LEDs to insects during the first half of the growing season (during the so-called "white nights"). Thank you once more, indeed.

Summary

Thank you very much once more for appreciating our manuscript entitled “Highly Improved Captures of the Diamondback Moth, Plutella xylostella, Using Bimodal Traps” (Manuscript ID: insects-3754389).

Reviewer 2 Report

Comments and Suggestions for Authors

This is a very interesting and overall well written paper. It shows very clearly that SSA and LED lights should be combined in DBM traps. I was a little confused by the statistical analyses in the paper and feel the authors should get some additional advice on how to analyze their data. Also, Figures 6 and 7 show additional data, but I could find no statistical analyses from these data. As mentioned below, please standardize the format for all of the figures in the paper. In general, numbers in the text ten and under need to be spelled out. This does not apply to usage, such as “Figures 6 and 7”. I also could not find statistical analyses for the quartile analyses in Table 3 or the multiplicity of the increase in DBM adult captures in Table 4. I found both of these data presentations unclear and, if kept, there needs to be  better explanations of how and why they were done.

General comments

Line 11 – “and the most destructive pest”. Also, change on line 37.

Line 20 – Change to “increase in adult insect attraction.”

Line 39 – The name kohlrabis should not be capitalized.

Line 40 – Please reword to “though there is no consensus of the place of DBM origin”

Lines 46-51 – I would suggest making this part of the previous paragraph and rewording the second sentence as follows, “For instance, at the end of the 20th century, annual losses from DBM were estimated at $1billion worldwide [11] and by the end of the first decade of the 21st century the harm caused by this insect is estimated to be in the range of $4-5 billion, according to the most conservative estimates [22].”

Lines 52-53 – “…DBM outbreaks has greatly increased over the past decade.”

Line 61 – replace “Anyway” with “Thus”.

Line 64 – Replace “most dangerous” with “worst”.

Line 70 – “by the interaction”.

Line 72 – “leads to breeding resistant DBM populations”.

Line 78 – “Replace “aims” with “includes”.

Lines 87-88 – “was discovered”.

Line 88 – Replace “became obvious” with was determined.

Line 89 – “and accordingly,”.

Line 104 – “an autodissemination process”.

Lines 114-115 – Replace “their most active flying activity” with “most of their flying activity”.

Line 116 – “adult movement is strongly”.

Lines 117-118 – The statement “it is almost instantly suppressed by light and activated almost immediately after dark” does not make sense if you are planning to trap them with light?  

Line 121 – The word microlepidoptera is not capitalized.

Line 124 – “incandescent lamps, LEDs”.

Line 127 – Change “low heating,” to “low heat output” and “mechanical damage, and low maintenance costs”.

Line 128 – Change “LEDs turned out to be able to’ to “LEDs are able to”.

Lines 135-136 – “(1) are easily placed in the habitats of target insect pests, due to their compact size, and (2)”.

Line 144 – “only a few articles”,

Line 147 – “effects of the interaction of these lures for capturing DBM adults”.

Figure 1 legend – “(C) General view of the site where the VIR world collection of cabbage was grown and where the traps for catching DBM adults were placed.

Line 165 – “VIR world collection of cabbage”.

Line 166 – “1,500 m2. The area had previously been planted in squash, pattypan, and pumpkin”.

Line 169 – “was grown with a seedling method”.

Line 170 – Do you mean the first week of June and not first decade of June? Also, spell out “first” and do not use “1st. Also, change “The number of plants per plot was 20” to “There were 20 plant per plot”.

Line 171 – Start the sentence with “Three”, instead of “3”.

Lines 177-178 – Change “after planting the seed-177lings 16 days later,” to “16 days after planting the seedlings”.

Line 190 – “we used Delta traps”.

Line 193 – “which contained a power supply”.

Line 194 – “a board with two low-“

Line 195 – “a control unit”.

Line 197 – Delete “and simply”.

Line 198 – “changed, if necessary.” Also, do you mean “Batteries were recharged using a solar panel mounted on one of the side faces of the trap”?

Line 209 – “discharge; and (6) determination”.

Line 211 – “that a LED power”.

Line 213 – Delete “, obviously,”.

Line 216 – “of a pulsed LED power supply”.

Lines 222-224 – “[a mix-222ture of (Z)-11-hexadecenal and (Z)-11-hexadecenyl acetate in a ratio of 10:90, to which as 223a minor component (0.99% of the total composition) (Z)-11-hexadecene-1-ol was added].”.

Line 225 – “To trap the attracted insects,”.

Lines 286-288 – Please briefly describe the magnitude of the differences. For instance, “SSA + LED traps captured almost three times more DBMs than LED alone and almost 13 times more than SSA alone”. Make sure my values are correct. Change “variant” to “treatment”.

Line 303 – Once again, what was the magnitude of the differences between male and female captures?

Table 2 – What does the asterisk stand for in “48.11 ± 14.14 a 1*”?. Please make it more clear which results are the controls.

Figure 6 – The y- and x-axes on the plots need labels. Also, most readers will not understand the labeling scheme for the weeks of the different bar groups. I have never seen this nomenclature before. Plot key needs to be larger and perhaps placed in the top right of the 2022 plot. Reword the figure legend as follows, “Figure 6. Dynamics of DBM adult (males + females) captures (xÌ…± SE) with the four different treatments over a weekly period from June 8 until September 14 in 2022-2024. SSA = synthetic sex attractant, LED = light emitting diodes, and Control is traps equipped with neither of these”. The control is not a bait. Make these changes to other figures.

Figure 7 – I think these plots need to be remade. The data bars are squeezed in the center and there are extra tick marks on the x-axis. Also, the y- and x-axes on the plots need labels. See the comments above for rewording the figure legend. It needs to be clearer that these are averages of three day collection periods throughout the study. The key is too small and does not include the treatments. These need to be added to the figure legend, as well. Finally, all of the figures in the paper need to be standardized to look similar. What do the error bars represent?

Where are the statistics for the data in Figures 6 and 7?

I do not understand the relevance of the data in Table 3

Line 363 – Change “catchability” to “catch”.

Table 5 – Please supply all of the P values for these correlations. Significant results cannot be highlighted and need to be in boldface.

Discussion

Line 415 – “various methods of attraction”.

Line 427 – “The joint functioning within traps of attraction tools with diverse modalities”.

Line 428-429 – It is unclear what the authors means by this phrase, “(1) asubstantial enhancement in the dependability of parent-offspring regression relationships as a predictive framework for monitoring models,”. It is not explained later in the discussion.

Line 437-438 – What is the context of the is statement here?: “even when virgin females were placed into a specially designed chamber within the light trap.”

Line 415 – “and (iii) possibly enhancing…”

Line 454 – “There is a large amount of evidence in the literature for the increased efficiency of using…”

Line 456-457 – “which achieves an increase in attraction”.

Lines 458-459 – “to increase the catch of sweet potato weevil males, Cylas formicarius (Fabr.), almost five-fold”.

Lines 459-465 – “In addition, traps equipped with UV-emitting lamps and SSA attracted six times more cabbage looper (Trichoplusia ni) males than traps with UV lamps alone and t 461the combination in traps of UV-LED with semiochemicals resulted in a 2-12-fold increase in catches of adults of four harmful Lepidoptera species, including orchard codling moth, Cydia pomonella (L.), oriental fruit moth, Grapholita molesta (Busck), oblique banded leafroller, Choristoneura rosaceana (Har.), and eye-spotted budmoth, Spilonota ocellana (Den. et Schiff.).”.

Line 466-467 – “Interestingly, when low-intensity LEDs were used alone for trapping, no insects were attracted”.

Line 495 – Seems that weather and climatic conditions are the same thing. I would delate one.

Line 500 – “Also, since pheromone and light traps are equipped…”

Line 509 – “exceptional additive interaction…”

Conclusion

Line 657 – “Trials of traps equipped with four treatments…”

Line 658 – “installed in cabbage fields…”.

Lines 660-662 – Replace “The trials were successful in demonstrating a very high level of synergy be-tween the SSA and UV LEDs in attracting DBM adults. The synergistic effect was assessed 660by a 15-fold increase in DBM adult (mostly represented by males) catches after retrofitting 661traps with SSA with a second bait as LED.” With “The trials were successful in demonstrating a very high level of synergy between SSA and UV LEDs in attracting DBM adults, resulting in a 15-fold increase in DBM adult (mostly represented by males) catches”.

Lines 663-666 – This sentence is unclear: “The high stability of statistically significant relationships between the number of trapped moths in traps with SSA and LED on one hand, and SSA+LED on the other, indicates the possibility of obtaining more balanced estimates of pest population dynamics using bimodal traps”.

Comments on the Quality of English Language

See my general comments.

Author Response

Author's Reply to the Review Report (Reviewer 2)

Comments and Suggestions for Authors

This is a very interesting and overall well written paper. It shows very clearly that SSA and LED lights should be combined in DBM traps. I was a little confused by the statistical analyses in the paper and feel the authors should get some additional advice on how to analyze their data. Also, Figures 6 and 7 show additional data, but I could find no statistical analyses from these data. As mentioned below, please standardize the format for all of the figures in the paper. In general, numbers in the text ten and under need to be spelled out. This does not apply to usage, such as “Figures 6 and 7”. I also could not find statistical analyses for the quartile analyses in Table 3 or the multiplicity of the increase in DBM adult captures in Table 4. I found both of these data presentations unclear and, if kept, there needs to be  better explanations of how and why they were done.

Response: Dear Reviewer 2,

Thank you very much for appreciating our work and for your very useful comments and advices. We tried to improve all parts of the article you commented on. We changed all criticized Figures, Tables, and their analysis. The article has been corrected according to your valuable suggestions and instructions (the text edited is written in red font). We would like to deeply thank the reviewer for their critical reading and helpful suggestions for improving our article. We have done our best to follow the suggestions and revise our article accordingly.

As for the statistical analysis, we have significantly improved it. Thus, Figures 6 and 7 formed the main part of Section 3.2. “Monotonic Relationship Among the Distributions of DBM Catches with Different Treatments in Traps”. In this section, using the Page criterion and the Friedman test, we analyzed the results of capturing DBM adults in order to check how unchanged the order of moth catch distributions in treatments remains over time. The link to the Page’s article has been added to the list of references: Page, E.B. Ordered hypotheses for multiple treatments: a significance test for linear ranks. J. Am. Stat. Assoc. 1963, 58, 216–230. So, together with the results of statistical analyses using the Kruskal-Wallis, Mood's Median и Adjusted Rank Transform tests presented in Section 3.1, evidence of monotony in the order of catch distributions presented in Section 3.2. provides a statistically sound basis for calculating the synergy effect when combining LED and SAA, which is presented in Section 3.3. ”The Increase in the Number of DBM Adults Collected by Traps Due to the Interaction of Bait Attractiveness".

As for Figures 6 and 7, they have been corrected in accordance with your valuable comments: their format has been standardized and the axes have been marked. In addition, Figure 7 was completely redrawn: (1) for uniformity, a recalculation was made based on weekly catches (as in Figure 6) instead of three-day ones; (2) unnecessary pictures describing the separate distributions of male and female catches were removed and only an image of their combined trapping distribution was left, since the analysis of female trapping was completed in section 3.1 because catches of females by traps were extremely weak.

The tables using quartiles to calculate additional estimates of synergy values were removed from the article, because you rightly pointed out that it was unclear why they were used. After all, in order to explain their use, additional data on the variation in natural illumination levels at high latitudes and its effect on LED attractiveness in traps needed to be introduced into the analysis, but this was beyond the scope of the study.

Next, according to your advice, the numbers "10" and "below" were converted to text in the text. Thanks a lot!

General comments

All of your comments listed below have been implemented. Thanks again so much for them!

Line 11 – “and the most destructive pest”. Also, change on line 37.  Done

Line 20 – Change to “increase in adult insect attraction.” Done

Line 39 – The name kohlrabis should not be capitalized. Done

Line 40 – Please reword to “though there is no consensus of the place of DBM origin” Done

Lines 46-51 – I would suggest making this part of the previous paragraph and rewording the second sentence as follows, “For instance, at the end of the 20th century, annual losses from DBM were estimated at $1billion worldwide [11] and by the end of the first decade of the 21st century the harm caused by this insect is estimated to be in the range of $4-5 billion, according to the most conservative estimates [22].” Done

Lines 52-53 – “…DBM outbreaks has greatly increased over the past decade.” Done

Line 61 – replace “Anyway” with “Thus”. Done

Line 64 – Replace “most dangerous” with “worst”. Done

Line 70 – “by the interaction”. Done

Line 72 – “leads to breeding resistant DBM populations”. Done

Line 78 – “Replace “aims” with “includes”. Done

Lines 87-88 – “was discovered”. Done

Line 88 – Replace “became obvious” with was determined. Done

Line 89 – “and accordingly,”. Done

Line 104 – “an autodissemination process”. Done

Lines 114-115 – Replace “their most active flying activity” with “most of their flying activity”. Done

Line 116 – “adult movement is strongly”. Done

Lines 117-118 – The statement “it is almost instantly suppressed by light and activated almost immediately after dark” does not make sense if you are planning to trap them with light?   Revised: “it is almost instantly suppressed by light simulating daylight, and activated almost immediately after darkness”

Line 121 – The word microlepidoptera is not capitalized. Done

Line 124 – “incandescent lamps, LEDs”. Done

Line 127 – Change “low heating,” to “low heat output” and “mechanical damage, and low maintenance costs”. Done

Line 128 – Change “LEDs turned out to be able to’ to “LEDs are able to”. Done

Lines 135-136 – “(1) are easily placed in the habitats of target insect pests, due to their compact size, and (2)”. Done

Line 144 – “only a few articles”, Done

Line 147 – “effects of the interaction of these lures for capturing DBM adults”. Done

Figure 1 legend – “(C) General view of the site where the VIR world collection of cabbage was grown and where the traps for catching DBM adults were placed. Done

Line 165 – “VIR world collection of cabbage”. Done

Line 166 – “1,500 m2. The area had previously been planted in squash, pattypan, and pumpkin”. Done

Line 169 – “was grown with a seedling method”. Done

Line 170 – Do you mean the first week of June and not first decade of June? Also, spell out “first” and do not use “1st. Also, change “The number of plants per plot was 20” to “There were 20 plant per plot”.  Yes, of course, first week of June, thank you! And, also “change” is done

Line 171 – Start the sentence with “Three”, instead of “3”. Done

Lines 177-178 – Change “after planting the seed-177lings 16 days later,” to “16 days after planting the seedlings”. Done

Line 190 – “we used Delta traps”. Done

Line 193 – “which contained a power supply”. Done

Line 194 – “a board with two low-“ Done

Line 195 – “a control unit”. Done

Line 197 – Delete “and simply”. Done

Line 198 – “changed, if necessary.” Also, do you mean “Batteries were recharged using a solar panel mounted on one of the side faces of the trap”? Yes, of course! Correction made

Line 209 – “discharge; and (6) determination”. Done

Line 211 – “that a LED power”. Done

Line 213 – Delete “, obviously,”. Done

Line 216 – “of a pulsed LED power supply”. Done

Lines 222-224 – “[a mix-222ture of (Z)-11-hexadecenal and (Z)-11-hexadecenyl acetate in a ratio of 10:90, to which as 223a minor component (0.99% of the total composition) (Z)-11-hexadecene-1-ol was added].”. Done

Line 225 – “To trap the attracted insects,”. Done

Lines 286-288 – Please briefly describe the magnitude of the differences. For instance, “SSA + LED traps captured almost three times more DBMs than LED alone and almost 13 times more than SSA alone”. Make sure my values are correct. Change “variant” to “treatment”. Oh, thanks, once more. Done

Line 303 – Once again, what was the magnitude of the differences between male and female captures? We add the text: “As for the females, the SSA did not appear to be attractive, as expected, while the LEDs were attractive, but approximately 55 times less than for males on average over 3 years of trials”.

Table 2 – What does the asterisk stand for in “48.11 ± 14.14 a 1*”?. Please make it more clear which results are the controls. Thank you, asterisk has been deleted

 Figure 6 – The y- and x-axes on the plots need labels. Also, most readers will not understand the labeling scheme for the weeks of the different bar groups. I have never seen this nomenclature before. Plot key needs to be larger and perhaps placed in the top right of the 2022 plot. Reword the figure legend as follows, “Figure 6. Dynamics of DBM adult (males + females) captures (xÌ…± SE) with the four different treatments over a weekly period from June 8 until September 14 in 2022-2024. SSA = synthetic sex attractant, LED = light emitting diodes, and Control is traps equipped with neither of these”. The control is not a bait. Make these changes to other figures. Thank you, all comments have been taken into account and corrections were made: (1) the y- and x-axes are provided with labels, (2) the legend is inserted, here and everywhere else we do not call the control a bait. As for the labeling scheme we used, the logic of its use is determined by using the Page and Friedman tests analyzing catch distributions (as mentioned above).

Figure 7 – I think these plots need to be remade. The data bars are squeezed in the center and there are extra tick marks on the x-axis. Also, the y- and x-axes on the plots need labels. See the comments above for rewording the figure legend. It needs to be clearer that these are averages of three day collection periods throughout the study. The key is too small and does not include the treatments. These need to be added to the figure legend, as well. Finally, all of the figures in the paper need to be standardized to look similar. What do the error bars represent? Thanks once more for the comment. It has already been mentioned above that the section of the article containing Figures 6 and 7 has been revised.

 Where are the statistics for the data in Figures 6 and 7? OK, it was already mentioned above that the section was supplemented with an analysis of the DBM adult catch distributions in time using the Page and Friedman tests.

I do not understand the relevance of the data in Table 3. OK, thank you, this table has been deleted

Line 363 – Change “catchability” to “catch”. Done

Table 5 – Please supply all of the P values for these correlations. Significant results cannot be highlighted and need to be in boldface. Completed, thank you. Table 5 has become quite decent.

Discussion

Line 415 – “various methods of attraction”. Done

Line 427 – “The joint functioning within traps of attraction tools with diverse modalities”. Done

Line 428-429 – It is unclear what the authors means by this phrase, “(1) a substantial enhancement in the dependability of parent-offspring regression relationships as a predictive framework for monitoring models,”. It is not explained later in the discussion. The phrase has been changed: (1) a substantial enhancement in the dependability of parent-offspring regression relationships as a predictive framework for monitoring models when low population occurs

Line 437-438 – What is the context of the is statement here?: “even when virgin females were placed into a specially designed chamber within the light trap.” The phrase has been changed: “This approach has proven successful in capturing males of various moths’ species, such as cabbage looper, Trichoplusia ni (Hbn.) [172,173], tobacco hawk moth, Manduca sexta (L.) [174–176], and tobacco budworm, Chloridea virescens (F.) [177], when virgin females were placed into a specially designed chamber within the light trap”.

Line 415 – “and (iii) possibly enhancing…” Done

Line 454 – “There is a large amount of evidence in the literature for the increased efficiency of using…” Done

Line 456-457 – “which achieves an increase in attraction”. Done

Lines 458-459 – “to increase the catch of sweet potato weevil males, Cylas formicarius (Fabr.), almost five-fold”. Done

Lines 459-465 – “In addition, traps equipped with UV-emitting lamps and SSA attracted six times more cabbage looper (Trichoplusia ni) males than traps with UV lamps alone and t 461the combination in traps of UV-LED with semiochemicals resulted in a 2-12-fold increase in catches of adults of four harmful Lepidoptera species, including orchard codling moth, Cydia pomonella (L.), oriental fruit moth, Grapholita molesta (Busck), oblique banded leafroller, Choristoneura rosaceana (Har.), and eye-spotted budmoth, Spilonota ocellana (Den. et Schiff.).”. Done

Line 466-467 – “Interestingly, when low-intensity LEDs were used alone for trapping, no insects were attracted”. Done

Line 495 – Seems that weather and climatic conditions are the same thing. I would delate one. Corrected: “weather or climatic conditions.”

Line 500 – “Also, since pheromone and light traps are equipped…” Done

Line 509 – “exceptional additive interaction…” Done

Conclusion

Line 657 – “Trials of traps equipped with four treatments…” Done

Line 658 – “installed in cabbage fields…”. Done

Lines 660-662 – Replace “The trials were successful in demonstrating a very high level of synergy be-tween the SSA and UV LEDs in attracting DBM adults. The synergistic effect was assessed 660by a 15-fold increase in DBM adult (mostly represented by males) catches after retrofitting 661traps with SSA with a second bait as LED.” With “The trials were successful in demonstrating a very high level of synergy between SSA and UV LEDs in attracting DBM adults, resulting in a 15-fold increase in DBM adult (mostly represented by males) catches”. Done

Lines 663-666 – This sentence is unclear: “The high stability of statistically significant relationships between the number of trapped moths in traps with SSA and LED on one hand, and SSA+LED on the other, indicates the possibility of obtaining more balanced estimates of pest population dynamics using bimodal traps”. The sentence was deleted. Thank you

Summary

Thank you very much once more for appreciating our manuscript entitled “Highly Improved Captures of the Diamondback Moth, Plutella xylostella, Using Bimodal Traps” (Manuscript ID: insects-3754389).

Round 2

Reviewer 2 Report

Comments and Suggestions for Authors

Dear authors. You did a great job making suggested changes and improving the manuscript!!!